# Fabrication of High-Sensitivity Optical Fiber Sensor by an Improved Arc-Discharge Heating System

**DOI:** 10.3390/s23063238

**Published:** 2023-03-18

**Authors:** Chao Ma, Jian Wang, Libo Yuan

**Affiliations:** 1Key Laboratory of In-Fiber Integrated Optics of Ministry of Education, College of Physics and Optoelectronic Engineering, Harbin Engineering University, Harbin 150001, China; 2Photonics Research Center, School of Optoelectronic Engineering, Guilin University of Electronic Technology, Guilin 541004, China

**Keywords:** arc-discharge method, dispersion turning point, helical long-period fiber grating, highly sensitive optical fiber sensor

## Abstract

We proposed a high-sensitivity optical fiber sensor based on a dual-resonance helical long-period fiber grating (HLPG). The grating is fabricated in a single-mode fiber (SMF) by using an improved arc-discharge heating system. The transmission spectra and the dual-resonance characteristics of the SMF-HLPG near the dispersion turning point (DTP) were studied through simulation. In the experiment, a four-electrode arc-discharge heating system was developed. The system can keep the surface temperature of optical fiber relatively constant during the grating preparation process, which shows an advantage in preparing high-quality triple- and single-helix HLPGs. In particular, benefiting from this manufacturing system, the SMF-HLPG operating near the DTP was successfully prepared directly by arc-discharge technology, without secondary processing of the grating. As a typical application example of the proposed SMF-HLPG, physical parameters such as temperature, torsion, curvature and strain can be measured with high sensitivity by monitoring the variation of the wavelength separation in the transmission spectrum. Therefore, the proposed sensor and its fabrication technology have potential application prospects in practical sensing measurement.

## 1. Introduction

Optical fiber sensors are playing an increasingly critical role in many practical applications, such as civil engineering, the automotive industry, aerospace and so on [1,2,3,4]. The sensor based on long-period fiber grating (LPFG) has shown great potential for small size, high accuracy, fast response and low cost [5,6,7]. Helical long-period fiber grating (HLPG) belongs to a new kind of LPFG and is characterized by its spiral refractive index modulation [8,9,10]. Because of its inherent spiral structure, the HLPG has shown many unique advantages in sensing measurement [11,12,13,14]. For example, it is possible to simultaneously measure the torsion direction and torsion rate. Note that in addition to functional specificity, high sensitivity is also a key factor of the sensors. Therefore, various schemes have been published for improving the response performance of the HLPG. Some strategies mainly include etching fiber cladding, using special fibers, combining functional materials and fiber tapering [15,16,17,18,19]. Moreover, it is also an attractive choice to use the coupling characteristics of the grating to realize high-sensitivity sensing. As reported, Shu et al. proved that the sensitivity of the LPFG can be significantly enhanced by selecting an appropriate period near the dispersion turning point (DTP), without using additional processing techniques [20]. In 2020, Ren et al. simulated and analyzed the turning point mode-coupling characteristics of the HLPG [21]. More recently, the HLPG working at the wavelength near the DTP has been proposed experimentally and proved to be highly sensitive to environmental parameters [22,23,24,25,26].

Generally, the HLPG near the DTP needs a short period to achieve mode coupling [27,28,29]. Therefore, the processing equipment used for HLPG preparation needs to have high precision and stability. At present, there are two main manufacturing technologies, namely CO_2_ laser beam heating technology and hydrogen–oxygen flame heating technology, which can directly realize the fabrication of HLPGs with turning point mode-coupling characteristics [22,26]. However, these preparation schemes are not perfect in the actual implementation process. For example, the high-power laser is very expensive, which increases the manufacturing cost. The hydrogen–oxygen generator is prone to producing redundant flammable and explosive gases while electrolyzing water. It is worth mentioning that the arc-discharge heating system is often used to prepare HLPGs because of its simplicity and security [30,31,32,33]. Unfortunately, the performance of this system is insufficient in terms of stability and repeatability, and it is difficult to achieve high-quality HLPG preparation [31,34]. Specifically, the fabrication of HLPG near the DTP has high requirements for manufacturing equipment. Because the slope of the phase-matching curve near the DTP region is the steepest, even small machining errors in fabrication will lead to poor energy coupling between modes. Therefore, in order to realize the preparation of target HLPG samples by arc-discharge technology, it is necessary to use expensive commercial fiber processing equipment and carry out post-processing on the HLPG [18]. In other words, it is still a challenge to directly fabricate the HLPG near the DTP with low cost and high efficiency by using arc-discharge technology.

In this paper, we developed an arc-discharge heating system with four electrodes, which can keep the surface temperature of optical fiber relatively constant during the grating preparation process. Because of its wide temperature distribution, the influence of fluctuation perpendicular to the fiber axis on the grating is weakened, so the system shows an advantage in preparing high-quality HLPGs. In particular, based on this promising fabrication system, the SMF-HLPG working near the DTP was successfully prepared directly by arc-discharge technology. The proposed SMF-HLPG exhibits two characteristic dips in the transmission spectrum based on the dual-resonance coupling mechanism. By monitoring the variation of wavelength separation, physical parameters such as temperature, torsion, bending and strain can be measured with high sensitivity. Therefore, the proposed method provides a stable and efficient way to realize the fabrication of highly sensitive HLPG sensors.

## 2. Theoretical Analysis

Relying on the axial periodic refractive index modulation of the fiber, the traditional LPFG can realize the energy exchange between the core mode and the co-propagating cladding modes. It is worth noting that, unlike the traditional LPFG, the HLPG is formed by twisting the optical fiber in the molten state, and the perturbation of its dielectric constant comes from the spiral distribution of the refractive index. In this paper, the SMF-HLPG was written in SMF with a single-helix structure. The corresponding configuration of the SMF-HLPG is shown in Figure 1. Since the transmitted light wave was mainly modulated by the spiral core, it can be considered that the SMF-HLPG was formed by twisting the optical fiber with an eccentric core. Therefore, in theoretical analysis, the SMF-HLPG is usually regarded as almost a standard optical fiber, except that its core follows the spiral path in the cladding [35,36,37].

The dual resonance characteristics of the SMF-HLPG near the DTP were simulated. The SMF used in this paper was provided by YOFC Inc., and the core and cladding diameters were 9 and 125 µm, respectively. This fiber had a core refractive index of 1.461 and a cladding refractive index of 1.457. By designing an appropriate spiral core structure in the fiber, it was possible to realize the coupling between the fundamental core mode and high-order cladding modes. The relationship between resonance wavelength and period can be determined by the phase-matching condition [38]:(1)$$\lambda_{res}=\left(n_{co}^{eff}-n_{cl}^{eff}\right)\Lambda \;,$$
where $n_{co}^{eff}$ and $n_{cl}^{eff}$ are the effective refractive indexes of the core mode and the cladding mode, respectively. In the simulation, the effective refractive indexes of fiber modes at different wavelengths were calculated. According to this equation, the phase-matching curves of four cladding modes (LP_1,8_, LP_1,9_, LP_1,10_, LP_1,11_) were obtained, as shown in Figure 2. It can be seen from the figure that the phase-matching curves of these high-order cladding modes had a non-monotonic variation trend. The slope of these curves changed the sign from positive to negative at a special point (red circles in the figure), which is called the DTP of the fiber mode. Obviously, if the period of the SMF-HLPG is slightly reduced from the DTP, one grating period will correspond to two resonance wavelengths. This is the dual-resonance coupling phenomenon near the DTP.

In order to further analyze the dual-resonance characteristics of the SMF-HLPG, a numerical model was established by using the Beam Propagation Method. According to the above analysis results, it can be found that the resonance wavelength at the DTP of LP_1,9_ is close to 1500 nm. Therefore, we analyzed the coupling characteristics between the core and the cladding modes in the period range of 250 µm to 270 µm. As typical examples, the SMF-HLPG with a period of 265 µm was simulated. For the purpose of ensuring simulation accuracy and calculation efficiency, the mesh sizes in the XY direction and Z direction were set to 0.5 µm. The energy distribution along the axis of the grating could be obtained by injecting a fundamental mode into the SMF-HLPG through the end face of the fiber. Figure 3a shows the optical field energy of the SMF-HLPG at a wavelength of 1360 nm. It can be observed that the core mode could be coupled with the cladding mode in the transmission. The energy of the core mode was monitored, and its distribution along the axial direction is shown in Figure 3b. When the coupling length reaches 28.9 mm (the coupling length corresponding to the orange star), most of energy of the core mode is transferred to the cladding mode.

Then, the energy distribution of the core mode in SMF-HLPG with the wavelength of 1.2–1.7 μm was calculated. As shown in Figure 4, there were two obvious loss regions in the simulation results, which means that the core mode could be coupled at two wavelengths. Subsequently, the transmission spectrum of the SMF-HLPG with a coupling length of 27.8 mm was obtained, as shown in the circle line in Figure 5. We can see that the transmission spectrum of the SMF-HLPG with a period of 265 µm had two obvious resonance dips. It was confirmed by simulation that these two resonance dips were caused by the coupling of core mode and LP_1,9_ cladding mode. The corresponding mode field distribution is shown in the inset of the figure. In addition, SMF-HLPGs with periods of 266 µm and 267 µm were simulated. The variation of the transmission spectra with different periods is presented in Figure 5. The red arrow indicates the moving direction of the resonance dip. We find that with the increase in the period, the two resonance dips in the transmission spectrum moved towards each other until they merged at a period of 267 µm. This phenomenon was consistent with the simulation results of the phase-matching curves.

In general, when the dual-resonance SMF-HLPG was disturbed by the environment, the waveguide dispersion characteristics of optical fiber played a key role in the evolution of the transmission spectrum. For example, as the grating is heated, the relationship between the resonance wavelength and the temperature can be expressed as [39,40]:(2)$$\frac{d\lambda_{res}}{dT}=\lambda_{res}\cdot \gamma \cdot \left(\alpha +\Gamma_{temp}\right)\;,$$
where *α* is the thermal expansion coefficient, Γ*_temp_* represents the temperature-dependent waveguide dispersion coefficient, *γ* describes the waveguide dispersion, which can be expressed as [41]:(3)$$\gamma =\frac{\frac{d\lambda_{res}}{d\Lambda }}{n_{co}^{neff}-n_{cl}^{neff}}\;,$$

As can be seen from the phase-matching curve in Figure 2, the curve in the area above the DTP had negative dispersion (*dλ_res_*/*d*Λ < 0), while the curve in the area below the DTP had positive dispersion (*dλ_res_*/*d*Λ > 0). Because the signs of dispersion were opposite, the two resonance dips in the transmission spectrum of the dual-resonance SMF-HLPG showed opposite moving trends when disturbed by external interference. Therefore, physical parameters could be measured with high sensitivity by monitoring the variation of the wavelength separation in the transmission spectrum.

## 3. Fabrication of the Dual-Resonance SMF-HLPG

An improved arc-discharge heating system was built to manufacture the SMF-HLPG which is schematically shown in Figure 6a. The mechanical devices in the proposed fabrication system mainly included a high-precision rotator, a fiber holder, a pulley, a pair of v-grooves, a translation platform and two pairs of electrodes. During preparation, one end of the SMF was fixed on the fiber holder. The other end of the fiber was installed along the central axis of the rotation motor and could be homogenously twisted at a constant speed of *V_1_*. Meanwhile, the SMF was directly heated into a molten state by the electric arc. Note that, unlike traditional arc-discharge equipment, here we used two pairs of electrodes to heat the fiber. The four electrodes were located in the same plane as the fiber and parallel to the workbench. In order to ensure electrical stability, the electrodes were fixed with mechanical fasteners in the preparation process, and the optical fiber moved along the axial direction with the translation platform at a speed of *V*_2_. In this way, we could obtain a SMF-HLPG with a period of Λ = 360°·*V*_2_/*V*_1_. In the preparation process, the surface temperature of optical fiber was recorded by an infrared thermal imager (ITI, A300, FLIR Systems Inc., Oregon, OR, USA) and displayed on the computer in real-time. Moreover, two v-grooves were employed to prevent the fiber from being far away from the uniform temperature zone due to vibration in the twisting process. In the system, the pulley and the weight were mainly used to keep the fiber straight between the electrodes. Figure 6b shows the details of the experimental setup.

Figure 7a shows an enlarged view of the electrodes. In the process of electrode discharge, a rectangular discharge area with a length of about 3 mm and a width of about 1.5 mm was formed, which could make the optical fiber heated more evenly and reduce the fluctuation of the unit length grating to some extent. The temperature performance of the heating source in the manufacturing system was one of the important factors affecting the grating quality. Figure 7b shows the surface temperature of the SMF during arc heating, indicating that the temperature distribution was relatively uniform. In addition, the change of the highest temperature on the fiber surface with time was also recorded, as shown in Figure 7c. The experimental results show that the temperature during the fiber heating process exceeded 1000 °C, with an average value of 1028 °C and a maximum fluctuation range of 48 °C. The standard deviation of the temperature was calculated to be 10.3 °C. This means that the four-electrode heating source used in the system was relatively stable and provided a high enough temperature field to completely melt the fiber.

Subsequently, the performance of the four-electrode arc-discharge heating system in the fabrication of HLPGs was tested. Firstly, we used this device to fabricate HLPGs in the triangular-core fiber (TCF, YOFC Inc., Wuhan, China). Figure 8a shows a microscopic image of the cross-section of the TCF. We can see that the cross-section of the TCF has a three-fold rotation symmetry. Then, the TCF-HLPG belongs to a triple-helix structure. Ten TCF-HLPG samples were obtained with the same preparation parameters. In the experiment, the period and coupling length were set to 810 µm and 17,010 µm, respectively. The microscopic image of the partial TCF-HLPG is shown in Figure 8b. It is obvious that the TCF-HLPG had an ultra-smooth surface with no visible deformation. The spectra of these ten TCF-HLPG samples were measured by using an optical spectrum analyzer (OSA, AQ6370C, YOKOGAWA, Tokyo, Japan) and a broadband light source (BBS). The measured spectra are shown in different colors in Figure 8c. These spectra were highly consistent. In particular, we found that the insertion loss at the non-resonant wavelength in these spectra was particularly small, which proves that the proposed system provides the possibility for realizing high-quality multi-helix HLPG preparation.

The changes in the main resonance dip (Dip 1) of these spectra were evaluated. Figure 9a shows the variation of resonance wavelength in the transmission spectra of different TCF-HLPG samples. The result shows that the maximum variation of the resonance wavelength was 3.65 nm. The variation of the transmission loss at the resonance wavelength in the transmission spectra of different TCF-HLPG samples is shown in Figure 9b. It can be seen that the maximum variation of transmission loss at the resonance wavelength was 4.37 dB. Experimental results indicated that the four-electrode arc-discharge heating system shows stable performance in fabricating triple-helix TCF-HLPGs with a highly consistent transmission spectrum.

Secondly, the ability of the heating system to prepare HLPGs in SMF (provided by YOFC Inc., Wuhan, China) was tested. Figure 10a shows a microscopic image of the cross-section of the SMF. The refractive index in the cross-section of conventional SMF is circularly symmetric. However, as mentioned in the previous chapter, the cross-section of the SMF-HLPG can be regarded as an optical fiber with an eccentric core due to the inevitable off-axis in the preparation process. Therefore, the SMF-HLPG is classified as a single-helix structure. Ten SMF-HLPG samples were prepared with the same period of 750 µm. The microscopic image of the partial SMF-HLPG is shown in Figure 10b. Similar to TCF-HLPG, the surface of SMF-HLPG was also very smooth, with no obvious deformation. The spectra of ten SMF-HLPG samples were measured and shown in different colors in Figure 10c. The results show that these spectra are also in good agreement. Therefore, this means that the transmission spectra of the single-helix SMF-HLPG samples prepared by the processing device can also achieve a high consistency.

The changes of Dip 1 in the transmission spectra of SMF-HLPGs were evaluated. The variation of resonance wavelength in the transmission spectra of different SMF-HLPG samples is shown in Figure 11a. As can be seen from the figure, the maximum variation of the resonance wavelength was 7.98 nm. Figure 11b shows the variation of the transmission loss at the resonance wavelength in the transmission spectra of different SMF-HLPG samples. We found that the maximum variation of transmission loss at the resonance wavelength was 3.65 dB. Therefore, the experimental results showed that the system also provides the possibility for preparing high-quality single-helix HLPG.

It is worth noting that, unlike the TCF-HLPG, the coupling lengths of SMF-HLPGs are not always the same in the experiment. This may be due to the different coupling mechanisms between the TCF-HLPG and the SMF-HLPG. It is reported that multi-helix HLPGs are produced by twisting a glass fiber with a non-circular core cross-section, while single-helix HLPGs are considered to be made by twisting a glass fiber with a non-concentric core [35,36,37]. In other words, the modulation of the refractive index of the TCF-HLPG comes from the coaxial twisted triangular fiber core. For the single-helix SMF-HLPG, the refractive index modulation mainly comes from the off-axis twisted fiber core. However, due to mechanical errors in the equipment, the offset of the fiber core is easily disturbed and changed during the SMF-HLPG preparation process. Therefore, the coupling lengths of the fabricated SMF-HLPG samples were not precisely identical. Figure 12 shows the coupling lengths of the TCF-HLPG and the SMF-HLPG samples in the experiment.

Finally, using the improved arc-discharge heating system, the SMF-HLPGs were prepared near the DTP. By adjusting the twist rate of the rotator and the speed of the translation platform, the grating period could be precisely tuned to the DTP. As shown in Figure 13, the SMF-HLPG samples were prepared by experiments with different periods of 256 μm, 257 µm and 258 µm. The corresponding coupling lengths were 26,880 µm, 30,583 μm and 26,316 µm, respectively. In the process of grating preparation, the twist rate of the rotator was set to 90°/s, and the speed of the translation platform was set to 64 µm/s, 64.25 µm/s and 64.5 µm/s, respectively. We found that for a given shorter grating period, the cladding mode can meet the phase-matching condition at two wavelengths, thus generating two resonance dips. As the period increased, the resonance dips of the transmission spectrum moved closer from opposite directions and eventually merged together to form a broadband resonance dip. The variation characteristics of these spectra were consistent with the above simulation results. There was a slight difference between the simulation and experiment with respect to the period, which was mainly caused by the thinning of fiber during grating preparation [25,42]. To further investigate the mode order of the resonance, the mode field image was captured by using a CCD (Model C12741-03, Hamamatsu Photonics, Shizuoka, Japan). As shown in the inset of Figure 13, this mode is identified as LP_1,9_.

## 4. Sensing Characteristics and Discussion

The sensing properties of the dual-resonance SMF-HLPG near the DTP were experimentally investigated. Firstly, the thermal response of the SMF-HLPG was studied. Employing a thermostat, the temperature of the SMF-HLPG was changed from 30 °C to 120 °C in a step of 10 °C. The variation of the transmission spectra with temperatures is shown in Figure 14a. The experimental result shows that with the increase in temperature, two resonance dips in the transmission spectrum had opposite responses and tended to be close to each other. This phenomenon is similar to the reported results of the LPFG near the DTP [20]. Therefore, it is further confirmed that the SMF-HLPG sample we prepared really worked near the DTP of the fiber mode. In the experiment, with the change of temperature, the variation of the wavelength separation between two resonance dips was measured three times. The corresponding experimental results are shown in Figure 14b. The mean wavelength separation of the dual-resonance SMF-HLPG varied monotonically, and its linear correlation coefficient and sensitivity were calculated to be 0.9923 and −306.96 pm/°C. Compared with the conventional SMF-HLPG made by the arc-discharge heating method, the thermal response of the dual-resonance SMF-HLPG proposed in this paper has been significantly improved [32].

The torsion characteristics of the dual-resonance SMF-HLPG were experimentally investigated by using a pair of concentric rotators. One of the rotators kept the fiber fixed, while the other could twist the fiber in a clockwise (CW) or counterclockwise (CCW) direction. In this experiment, the distance between the rotators was 21 cm, and the rotation angle of the optical fiber changed from −360° to 360° in 60° steps. The transmission spectra of the dual-resonance SMF-HLPG with different rotation angles are shown in Figure 15a. When the SMF-HLPG was subjected to torsional stress in the counterclockwise direction, the two resonance dips moved toward each other. However, when the SMF-HLPG was under clockwise mechanical torsion, these dips moved in opposite directions. This is mainly because the helical period of the SMF-HLPG was changed during the twisting process [13]. For example, the co-direction torsional stress can reduce the period of the SMF-HLPG and make the grating far away from the DTP, resulting in the reverse drift of the resonance dips in the spectrum. In the same way, the contra-direction torsional stress can enlarge the grating period, leading to the opposite situation. According to this principle, the proposed SMF-HLPG can be used to measure the torsion direction. In addition, the torsional response of the dual-resonance SMF-HLPG was quantitatively characterized. The dependence of wavelength separation of the two dips on the twist rate is shown in Figure 15b. The corresponding linear correlation coefficient was 0.9915. The arrows in the figure indicate the twist directions of the grating. The calculated torsion sensitivity was 209.14 pm/(rad/m), which is about twice as high as that of the conventional SMF-HLPG we published earlier [31].

Figure 16a illustrates the evolution of the transmission spectrum with curvature. In the experiment, the SMF-HLPG was fixed on a steel ruler, and its bending characteristics were measured by changing the curvature of the steel ruler from 0 m^−1^ to 1 m^−1^. With the increase in curvature, the distance between the two resonance dips in the transmission spectrum decreased gradually. Figure 16b shows the relationship between the wavelength separation and curvature. We can see that within the curvature range of 0–0.51 m^−1^, the wavelength separation between two resonance dips changes relatively slowly. However, the wavelength separation decreased significantly with the increase in curvature. A similar phenomenon was also found in the previous reports [15,43]. The measured maximum bending sensitivity of the proposed SMF-HLPG was −16.81 nm/m^−1^, which is almost the same as the results reported in previous studies [15,43].

The axial strain properties of the dual-resonance SMF-HLPG were studied experimentally. In the measurement, one end of the SMF-HLPG was held stationary by a fiber holder, while the other end was attached to a translation stage to provide the strain from 0 µε to 1000 µε. The transmission spectra of the dual-resonance SMF-HLPG were recorded at a strain interval of 100 µε, as illustrated in Figure 17a. Experimental results proved that with the increase in mechanical axial strain, the two resonance dips in the transmission spectrum approached each other. This is mainly due to the axial stretching of the SMF-HLPG leading to a longer period and making the resonance wavelength closer to the DTP of the fiber mode. The linear relationship between the wavelength separation and axial strain is presented in Figure 17b. The measured strain sensitivity was −10.33 nm/mε, which is about five times as high as that of the conventional SMF-HLPG made by the arc-discharge heating method [32].

In order to further evaluate the sensing performance of the dual-resonance SMF-HLPG proposed in this paper, Table 1 gives a comparison with the HLPGs made by the arc-discharge heating method. Obviously, compared with the HLPGs far away from the DTP, the sensitivity of the proposed dual-resonance SMF-HLPG to temperature, torsion, bending and strain has been significantly enhanced. In particular, this grating can be directly prepared by our proposed arc-discharge equipment without additional processing technology. It provides a stable and efficient way to realize the fabrication of highly sensitive HLPG sensors.

## 5. Conclusions

In summary, we have demonstrated a low-cost and high-efficiency approach to fabricating the high-sensitivity SMF-HLPG sensor. The four-electrode arc-discharge heating system developed in this paper can keep the surface temperature of optical fiber relatively constant during the grating preparation process. Experimental results show that the system has the advantage of preparing high-quality triple- and single-helix HLPGs. Employing the promising heating system, the SMF-HLPG working near the DTP was successfully directly manufactured by arc-discharge technology. The dual-resonance characteristics of the proposed SMF-HLPG were experimentally proved, which is in good agreement with the simulation results. In addition, by monitoring the variation of wavelength separation in the transmission spectrum, physical parameters such as temperature, torsion, bending and strain can be measured with high sensitivity. Therefore, we believe that the proposed sensor and its fabrication technology may have some potential applications in the field of optical fiber sensing.

## Figures and Tables

**Figure 1 sensors-23-03238-f001:**
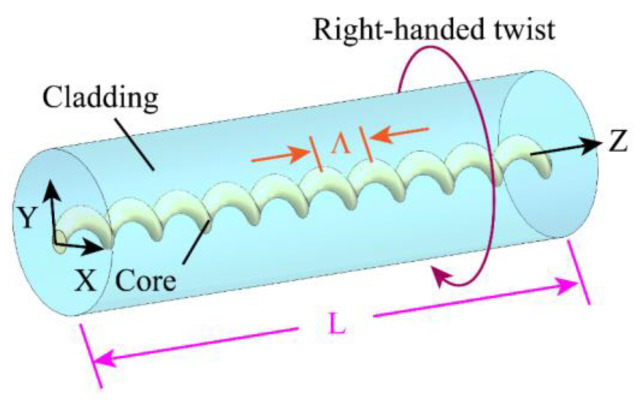
The corresponding configuration of the SMF-HLPG.

**Figure 2 sensors-23-03238-f002:**
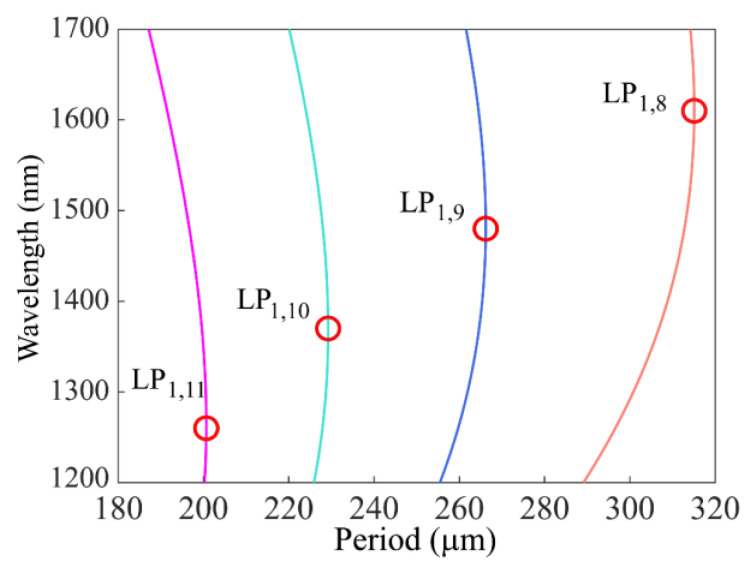
The relationship between resonance wavelength and grating period of four cladding modes.

**Figure 3 sensors-23-03238-f003:**
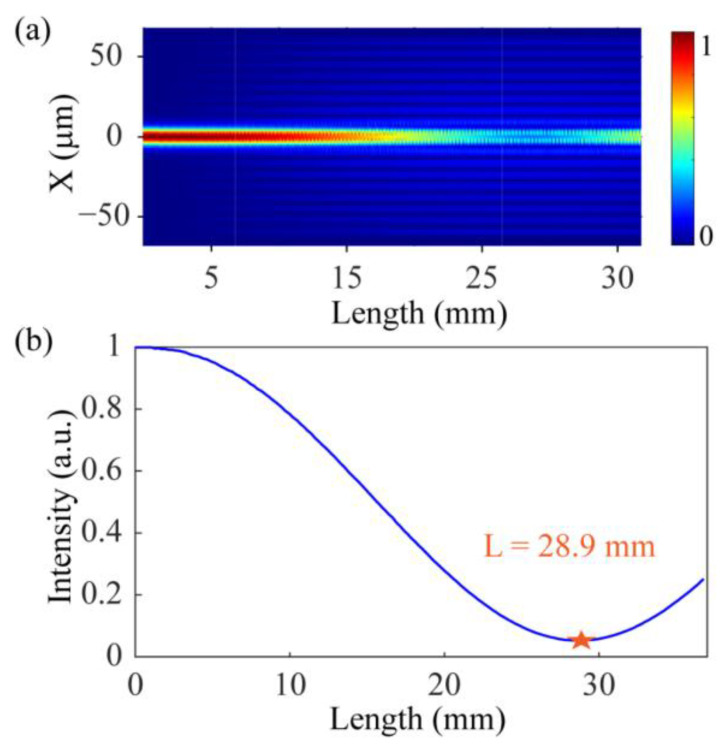
(**a**) Simulated transmission optical field distribution of the SMF-HLPG. (**b**) The light intensity of fundamental mode along the SMF-HLPG.

**Figure 4 sensors-23-03238-f004:**
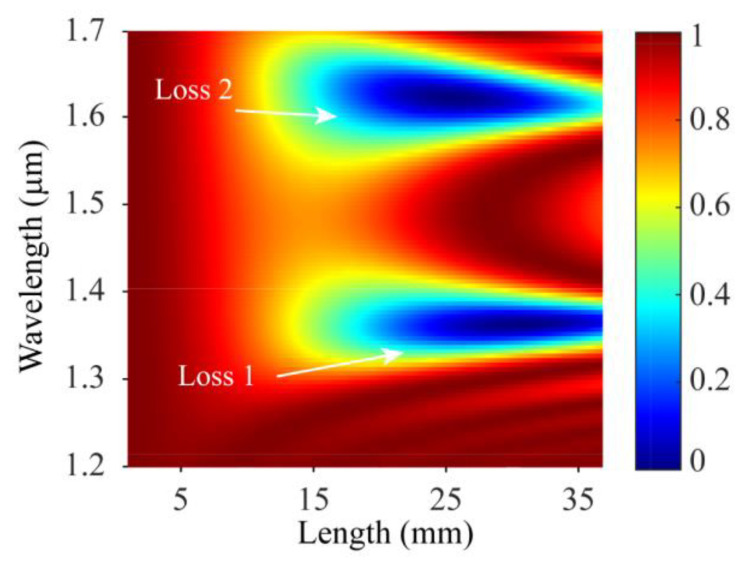
The energy distribution of the core mode of SMF-HLPG in the wavelength range of 1.2–1.7 μm.

**Figure 5 sensors-23-03238-f005:**
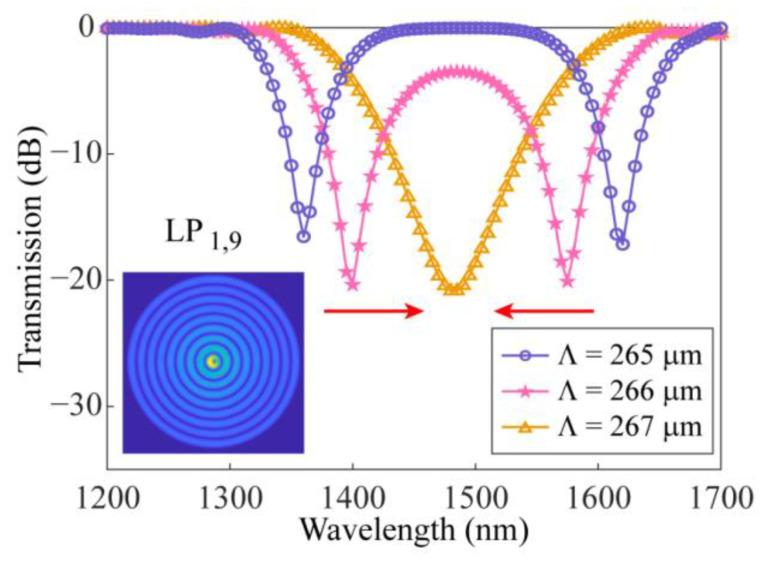
The simulated transmission spectra of the SMF-HLPGs with different grating periods.

**Figure 6 sensors-23-03238-f006:**
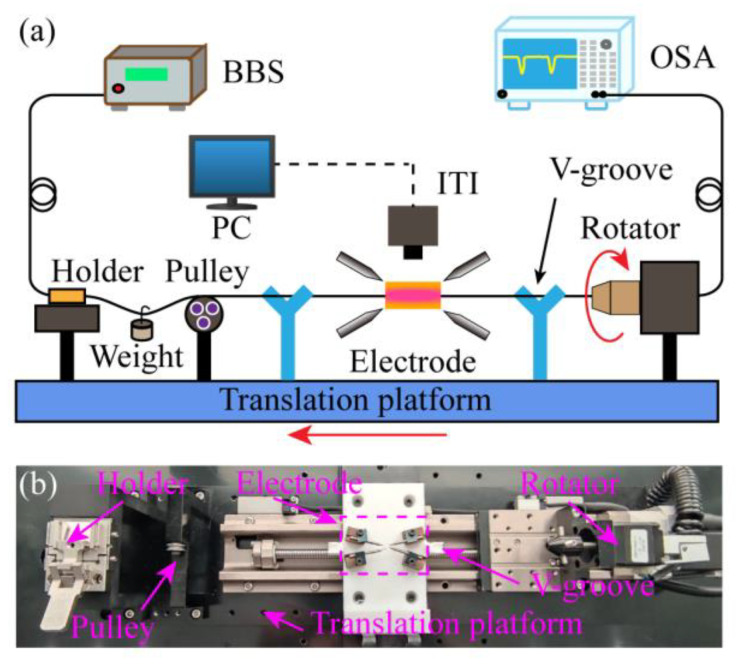
(**a**) The schematic diagram of the improved arc-discharge heating system. (**b**) The photograph of the four-electrode arc-discharge heating system.

**Figure 7 sensors-23-03238-f007:**
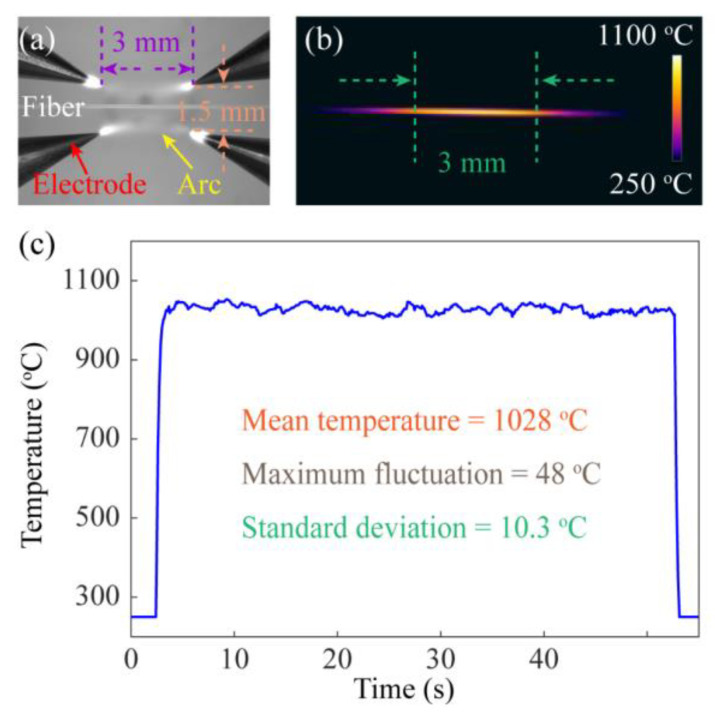
(**a**) The enlarged view of the electrodes. (**b**) Temperature distribution on the surface of the fiber during the arc discharge. (**c**) The variation of the highest temperature on the fiber surface with time.

**Figure 8 sensors-23-03238-f008:**
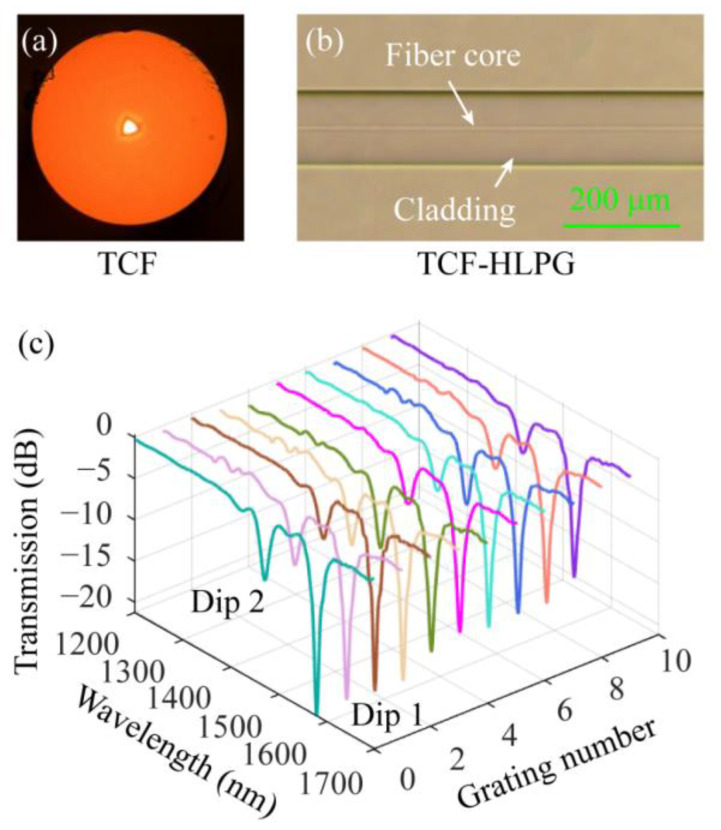
(**a**) Microscopic image of the cross-section of the TCF. (**b**) The microscopic image of the partial TCF-HLPG. (**c**) Transmission spectra of different TCF-HLPG samples.

**Figure 9 sensors-23-03238-f009:**
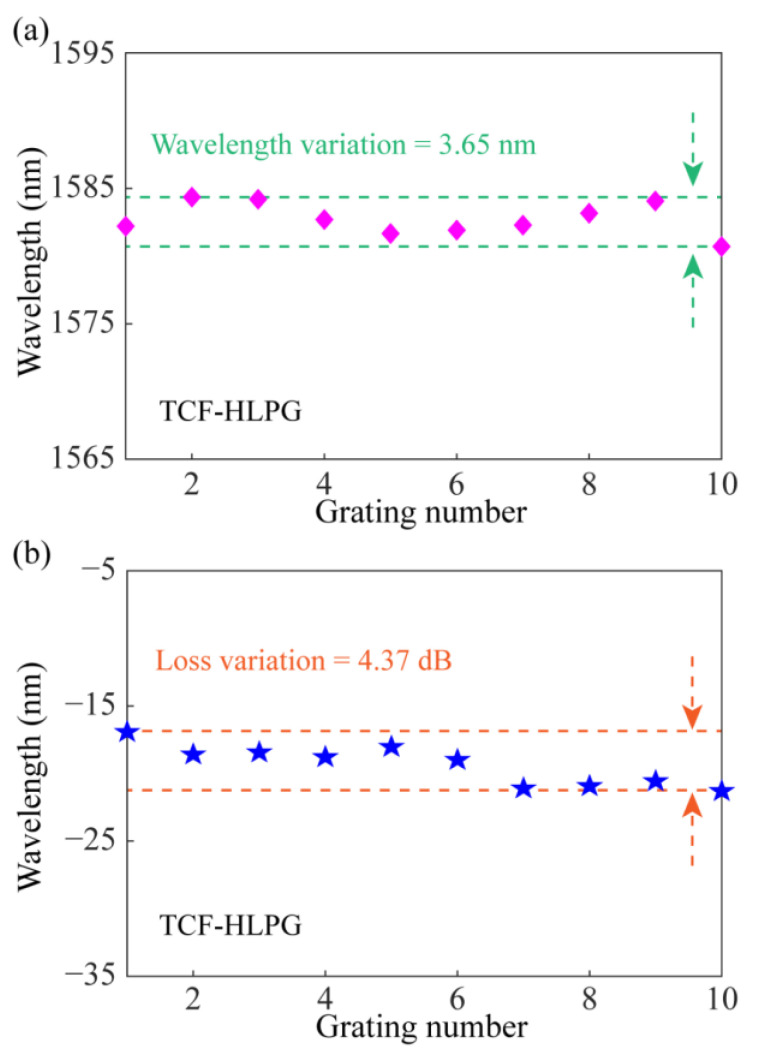
(**a**) The variation of resonance wavelength in the transmission spectra of different TCF-HLPG samples. (**b**) The variation of the transmission loss at the resonance wavelength in the spectra of different TCF-HLPG samples.

**Figure 10 sensors-23-03238-f010:**
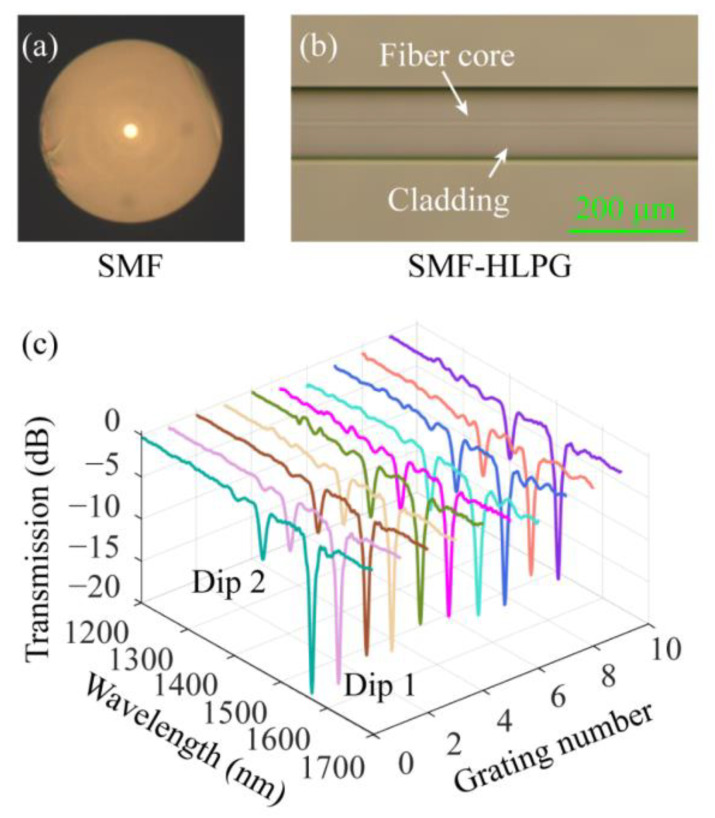
(**a**) Microscopic image of the cross-section of the SMF. (**b**) The microscopic image of the partial SMF-HLPG. (**c**) The transmission spectra of different SMF-HLPG samples.

**Figure 11 sensors-23-03238-f011:**
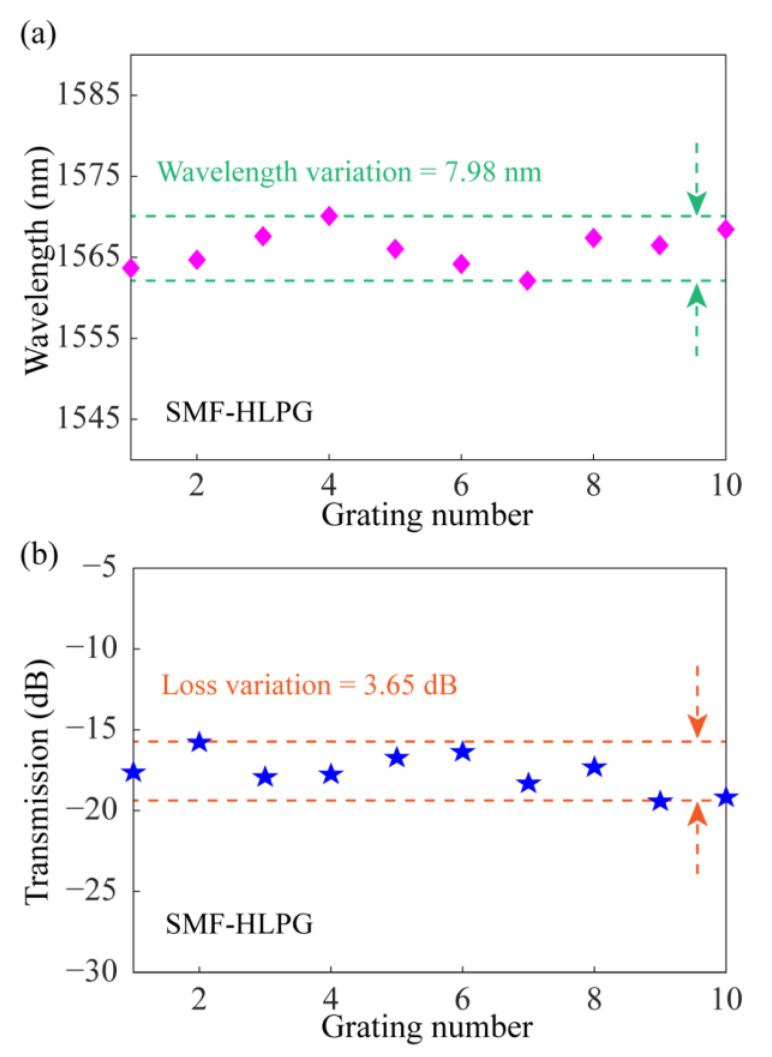
(**a**) The variation of resonance wavelength in the transmission spectra of different SMF-HLPG samples. (**b**) The variation of the transmission loss at the resonance wavelength in the spectra of different SMF-HLPG samples.

**Figure 12 sensors-23-03238-f012:**
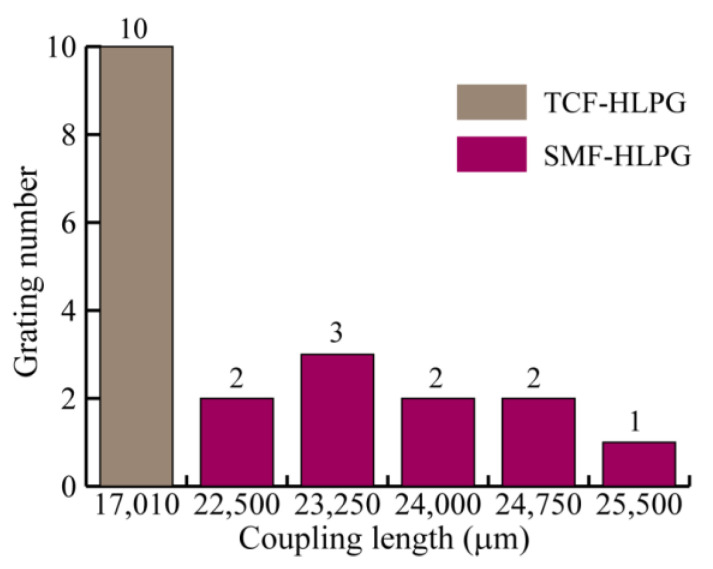
The coupling lengths of the TCF-HLPG and the SMF-HLPG samples.

**Figure 13 sensors-23-03238-f013:**
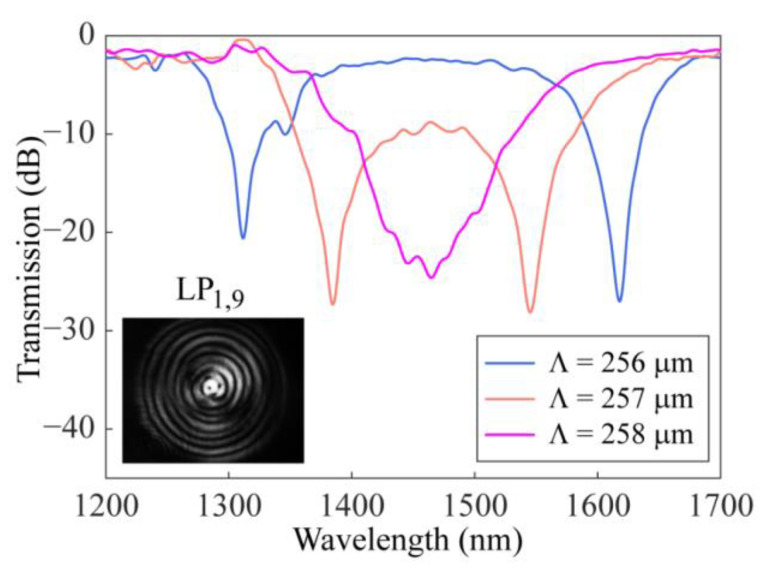
The experimental transmission spectra of the SMF-HLPGs with different grating periods. Inset shows the mode field distribution.

**Figure 14 sensors-23-03238-f014:**
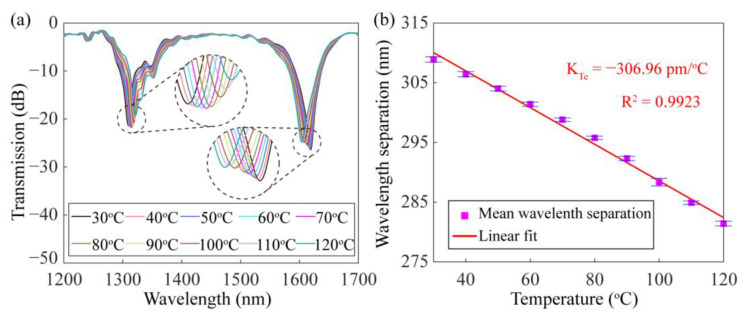
(**a**) Transmission spectra of the dual-resonance SMF-HLPG with different temperatures. (**b**) The dependence of mean wavelength separation on the temperature.

**Figure 15 sensors-23-03238-f015:**
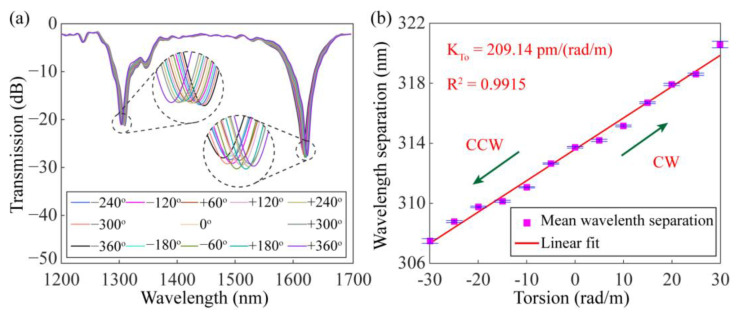
(**a**) Transmission spectra of the dual-resonance SMF-HLPG with different twist angles. (**b**) The dependence of mean wavelength separation on the torsion.

**Figure 16 sensors-23-03238-f016:**
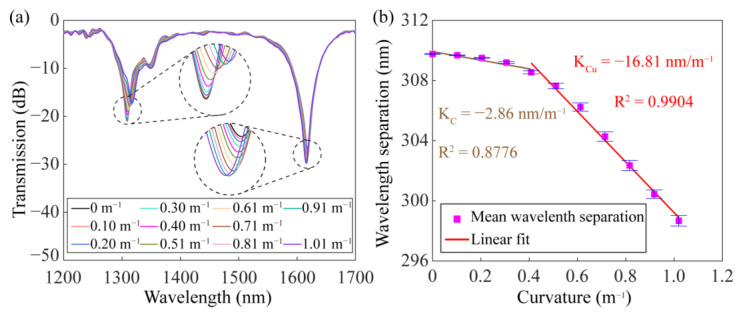
(**a**) Transmission spectra of the dual-resonance SMF-HLPG with different curvatures. (**b**) The dependence of mean wavelength separation on the curvature.

**Figure 17 sensors-23-03238-f017:**
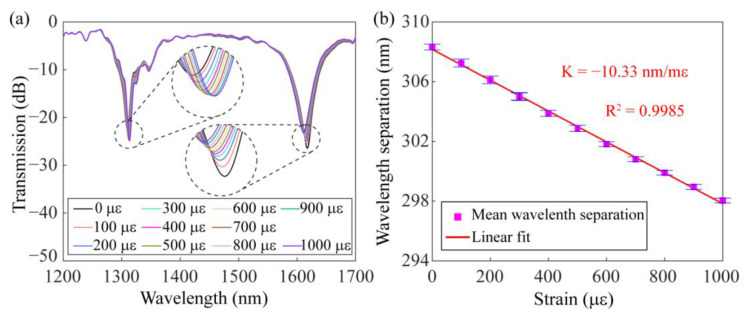
(**a**) Transmission spectra of the dual-resonance SMF-HLPG with different strains. (**b**) The dependence of mean wavelength separation on the strain.

**Table 1 sensors-23-03238-t001:** Comparison of the dual-resonance SMF-HLPG with different studies.

Structure	Temperature (pm/°C)	Torsion (pm/(rad/m))	Curvature (nm/m^−1^)	Strain (nm/mε)	Near DTP	APT. ^1^	Ref. ^2^
PBF-HLPG	10.5	115.5	–	1.84	No	No	[30]
SMF-HLPG	75	127.8	−6.77	−1.4	No	No	[31]
SMF-HLPG	70	−46.46	–	1.88	No	No	[32]
THF-HLPG	48.85	−183.85	−11.24	−2.6	No	No	[34]
SMF-HLPG	177.51	−321.57	–	−1.27	Yes	Yes	[18]
SMF-HLPG	306.96	209.14	−16.81	−10.33	Yes	No	This work

^1^ Additional processing technology. ^2^ Reference.

## Data Availability

Not applicable.

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
