# Peer review of "Fabrication of High-Sensitivity Optical Fiber Sensor by an Improved Arc-Discharge Heating System"

_sensors, 2023, doi:10.3390/s23063238_

Round 1

Reviewer 1 Report

This paper provided a novel method of fabricating HLPG sensors. Theoretical analysis and experiments with dual resonance also have been demonstrated.  The paper is interesting and instructive. It is suitable for this journal.

However, there is a problem need to be clarified.

The repeatability of the sensors manufacturing method needs to be verified by providing at least two sensors' data.

Reviewer 2 Report

In the manuscript (sensors-2257595), a low-cost and high-efficiency approach that employing four-electrode arc-discharge method to fabricating the high-sensitivity SMF-HLPG sensor was demonstrated. The fabrication method had been demonstrated with high stability and good repeatability by experiments. Moreover, the HLPG working near the DTP was studied in simulations and experiments. And the sensor show high sensitivities in temperature, torsion, bending, and strain detection. In my opinion, the manuscript can be accepted for publication after the following minor issues addressed.

(1)       From Fig. 3(b), the distance between electrodes shows about several hundreds of micrometers. Whereas, it is stated a 3mm*2.5mm rectangular temperature zone formed by electrodes discharge, and this zone is much larger than that enclosed by electrodes. Please make comments on this.

(2)       The scale bars should be added to Figure 3(b) and the inset in Figure 4(a).

(3)       What is the total length of the fabricated HLPG? Is an eccentric core SMF is used to fabricate the HLPG? Please give more information on the used SMF.

Reviewer 3 Report

Dear Authors, 

here are my comments:

1. The novelty in this paper is not shown in sufficient form. You developed the four electrodes method but why this method is better? You wrote "which has the characteristics of low cost, high stability and good repeatability". How did you investigated the stability and repeatability? 

2. In "Theoretical Analysis" chapter you presented only the figures. How did you obtain it? The full description of the numerical model must be presented. 

3.  The arc methods are easy to implement, but from my experience stability and good repeatability is very hard to achieve. So:

a)  You wrote: "The experimental results show that the temperature during the fiber heating process exceeds 1000 oC, with an 158 average value of 1028 oC and a maximum fluctuation range of 48 oC." How did you measure the temperature? What is uncertainty of this measurement? From Fig. 4a the initial temperature was 200-250 deg.C. How did you obtain such high initial temperature?

b) How many samples did you fabricate? What was the geometry of the samples? What was the repeatability of the geometry parameters of the samples and how did you investigate it? For example how did you investigate the period of samples? What is uncertainty of this measurement and also how did you investigate the repeatability?

4. You also presented the application of SMF-HLPG as temperature, torsion, curvanture and strain sensor. In my opinion the theoretical analysis of such implementation should be also added in order to enable veryfication the obtained results. 

Round 2

Reviewer 3 Report

Thank you for your work. So if you wrote "we did not study the change in optical fiber geometry" so how do you know exact value of the period of fabricated samples (Fig. 13) + line 287?   

Reviewer 4 Report

Thank you for the accurate responses. The article can be considered for publication.
